# A Novel de Novo Variant in 5′ UTR of the *NIPBL* Associated with Cornelia de Lange Syndrome

**DOI:** 10.3390/genes13050740

**Published:** 2022-04-22

**Authors:** Yonghua Chen, Qingqing Chen, Ke Yuan, Jianfang Zhu, Yanlan Fang, Qingfeng Yan, Chunlin Wang

**Affiliations:** 1Department of Pediatrics, The First Affiliated Hospital, Zhejiang University School of Medicine, Hangzhou 310003, China; 21518037@zju.edu.cn (Y.C.); chelena@zju.edu.cn (Q.C.); 21618001@zju.edu.cn (K.Y.); 6507932@zju.edu.cn (J.Z.); fangyanlan@zju.edu.cn (Y.F.); 2College of Life Science, Zhejiang University, Hangzhou 310027, China; qfyan@zju.edu.cn

**Keywords:** Cornelia de Lange syndrome, *NIPBL*, 5′ UTR, transcription

## Abstract

Background: Cornelia de Lange syndrome (CdLS) is a genetic syndrome characterized by intellectual disability, special facial features, growth retardation, feeding difficulties, and multiple organ system abnormalities. *NIPBL* variants occur in approximately 80% of CdLS cases. Aims: We report a novel de novo heterozygous pathogenic variant in the *NIPBL* and its association with CdLS. We also examined the key regulatory sequences of the 5′ untranslated region in *NIPBL* mRNA. Few studies have reported mutation sites in the 5′ untranslated region (UTR) of the *NIPBL* that result in CdLS. Methods: The patient’s medical history, clinical manifestations, physical examination, laboratory examination, Griffiths development assessment scale—Chinese version, and cardiac B-ultrasound were examined. Mutation screening was conducted using trio whole exome sequencing (trio-WES) and Sanger sequencing. Quantitative PCR was performed to measure the *NIPBL* expression in peripheral blood mononuclear cells. A Dual-Luciferase reporter assay was conducted to evaluate the transcription of truncated mutants. Results: The proband showed characteristics of CdLS including thick eyebrows, a concave nasal ridge, long and smooth philtrum, downturned corners of the mouth, intellectual disability, postnatal growth retardation, and a short fifth toe. A novel de novo heterozygous pathogenic variant in the *NIPBL* (c.-467C > T) was identified. A Dual-Luciferase reporter gene assay showed that SPO1 (-490 bp to -360 bp) and SPO3 (-490 bp to -401 bp) induced the highest activity. Conclusions: We found a novel de novo heterozygous pathogenic variant (c.-467C > T) in the *NIPBL* resulting in CdLS. Our findings expand the spectrum of pathogenic mutations for CdLS. Our in vitro experiments elucidated important regulatory sequences in the 5′ UTR of the *NIPBL*.

## 1. Introduction

Cornelia de Lange syndrome (CdLS MIM 122470, 300590, 610759, 614701, 300882) is a rare genetically heterogeneous disorder that affects multiple systems. It is characterized by prenatal and postnatal growth retardation, hirsutism, and intellectual disability. Patients with CdLS show distinctive facial features, such as microcephaly, synophrys, highly arched eyebrows, long eyelashes, short noses with anteverted nares, and small widely spaced teeth [1]. The incidence of CdLS ranges from between 1 in 10,000 to 1 in 30,000 live births [2]. CdLS is often identified at the time of birth according to the phenotype of the newborn, and only 23% patients are confirmed by prenatal diagnosis [3,4]. Seven genes (*NIPBL*, *SMC1A*, *SMC3*, *RAD21*, *BRD4*, *HDAC8,* and *ANKRD11*) have been linked to CdLS [5]. Among these genes, the *NIPBL* (MIM 122470) variants account for 80% of cases [6]. The inheritance pattern of CdLS is autosomal dominance or X-linked dominance, depending on the mutated gene.

The *NIPBL* gene, located on 5p13.2, encodes a cohesin loading factor that plays an essential role in cohesin binding to chromatin [7]. The process of genetic information transmission depends on the activity of the cohesin complex, which ensures genomic stability during cell division and DNA damage repair, and participates in the three-dimensional tissue of cell nuclear staining fibers [8,9,10]. Evidence suggests that cohesin, and the *NIPBL* proteins that regulate cohesin loading onto DNA, play a key role in gene regulation [11,12,13]. In addition, many studies have revealed that the aberrant expression of the *NIPBL* leads to abnormal development of the heart, limbs, and nervous system [14,15]. In vivo studies have demonstrated that the *NIPBL* is involved in craniofacial development by disturbing the function of MAU2 and interfering with the expression of *HOX*, resulting in limb outgrowth [16,17].

The *NIPBL* gene contains 47 exons and the translation codon is located in exon 2 instead of exon 1 [18]. Most of the mutations in the *NIPBL* that are associated with CdLS1 occur in the coding sequence [6,19]. The mutation loci in the non-coding sequence can be easily missed due to the struggle to capture it by whole exome sequencing. Moreover, few studies have reported mutation sites in the 5′ untranslated region (UTR) of the *NIPBL* that result in CdLS1 [18,20]. Therefore, it is clear that the pathogenic sequence in non-coding regions of the *NIPBL* is important for CdLS1 diagnosis.

In this study, we performed trio whole exome sequencing (trio-WES) on a patient with CdLS1 and identified a novel de novo heterozygous pathogenic variant in the 5′ UTR of the *NIPBL*. We confirmed that the mutation altered the transcription of the *NIPBL* and performed reporter gene assays of truncation mutants to identify vital transcription-regulating sequences in the 5′ UTR.

## 2. Materials and Methods

### 2.1. Subjects

The study protocol was approved by the Ethics Committee of The First Affiliated Hospital, Zhejiang University, China (approval number No. 2018-728). This study was conducted in accordance with the principles of the Declaration of Helsinki. Informed written consent was obtained from the parents of the proband for publication of this case.

### 2.2. Clinical Evaluations

We performed a detailed clinical assessment of and obtained clinical data about the proband, including age, birth history (antenatal, birth complication, gestation, and birth weight), and childhood characteristics. We also investigated laboratory findings including growth hormone (GH) stimulation tests with clonidine and arginine, and hematological and chemical profiles, in addition to conventional cytogenetic analysis, abdominal ultrasonography, echocardiography, and a Griffiths development assessment scale—Chinese version.

### 2.3. Cytogenetic and Molecular Studies

Peripheral blood samples from the proband and his parents were obtained under the premise of informed consent, and genomic DNA was isolated using a column extraction kit (Kang for the century Biotechnology Co., Ltd., Beijing, China.). Purified DNA (50 ng) underwent a process of end repair, phosphorylation, and ligation to barcoded sequencing adapters. Then, enriched target fragments and a constructed whole exome library were performed using a xGen^®^ Exome Research Panel v1.0 (Integrated DNA Technologies, IDT, Coralville, IA, USA) with capturing probes and liquid hybridization. Of these, the target DNA fragments covered 19,119 genes with whole exons and partial introns and each enriched region shared 40 Mb of targeted sequences. High-throughput sequencing was performed by an Illumina NovaSeq 6000 series sequencer (PE150), and then clean data from the raw sequencing data was processed with quality filtered, such as removing PCR duplicates and low-quality reads. Variant calling was generated by aligning the clean data to the NCBI human reference genome (hg18). Subsequently, Samtools and Pindel were used to confirm SNPs (Single Nucleotide Polymorphisms) and indels, respectively. Variability classification was completed by using a three-factor system and the American College of Medical Genetics (ACMG) genetic variation grading system. Finally, nonsynonymous substitutions and SNPs with a MAF (minor allele frequency) lower than 5% were filtered using SIFT. Meanwhile, the function of mutated genes and their pathogenicity were then analyzed by referencing to dbSNP, OMIM, Swiss-var, HGMD, ClinVar, and other disease databases. The candidate causal genes discovered by trio-WES were confirmed by Sanger sequencing. Chromas Lite v2.01 (Technelysium Pty Ltd., Tewantin, QLD, Australia) was used for the Sanger sequencing. Sequences of primers used to amplify the affected fragment of the *NIPBL* were as follows: forward, 5′-TACACCCGGCCGGAGAACCTAAAA-3′; and reverse5′ ATGCATCGAGCTGAAAACCGAAAA-3′.

### 2.4. Quantitative Real-Time PCR

Ficoll^®^ PM 400 (Sigma-Aldrich, St. Louis, MO, USA) was used to isolate monocytes from fresh peripheral blood. Total RNA was extracted from monocytes using a Trizol reagent (Invitrogen). The amount of RNA extracted was quantified and reverse-transcribed into cDNA by PrimeScript™ Reverse Transcriptase (Takara, Shiga Prefecture, Japan), following the manufacturer’s instructions. qRT-PCR was performed using SYBR Green qPCR Mastermix (Thermo Fisher Scientific, Waltham, MA, USA) with the *NIPBL* primers (forward, 5′-AGATGCAACACATCGGTATC-3′; reverse, 5′-GAATCTCCATCGTCACTACTTAG-3′). The Ct values of the *NIPBL* were normalized to levels of the house-keeping GADPH mRNA (forward, 5′-AGGTTCTTCCCGCTCTCAAT-3′; reverse, 5′-CCTCCTTGATAGCAGCCTTG-3′).

### 2.5. Plasmid Construction and Dual-Luciferase Assays

The 5′ UTR region (489 bp (c.-490 to c.-1) of the *NIPBL*) was amplified by PCR using cDNA extracted from HEK293T cells as a template and the following primers: forward 5′-ATTTTGTTCTGAGAGGGAG-3′ and reverse 5′-CCTGAATTTCTGGAATGG-3′. The pGL4.48 [luc2] vector (containing a minimal promoter and firefly luciferase sequence) and the pGL4.74 [hRluc/TK] vector (containing the Renilla luciferase gene) were purchased from Promega (USA). The *NIPBL* 5′ UTR region was cloned into the pGL4.48 luciferase reporter plasmid. The full-length 5′ UTR fragment was named SPO0. We also constructed a series of truncated 5′ UTR sequences named as SPO1 (c.-490 to c.-360), SPO2 (c.-360 to c.-1), SPO3 (c.-490 to c.-401), SPO4 (c.-401 to -360), SPO5 (c.-490 to c.-441), and SPO6 (c.-441 to c.-401), respectively, (ATG start codon = +1). 

HEK293T cells were plated into 24-well plates at a confluency of 80–90% before transient transfection. The reporter plasmids and helper plasmid pGL4.74 [hRluc/TK] were co-transfected at a ratio of 50:1 (total of 500 ng DNA) using Lipofectamine™ 3000 transfection reagent (Thermo Fisher, Waltham, MA, USA). The culture medium was replaced with fresh medium 6–8 h later.

At 12 h after transfection, cells were seeded into 96-well plates at a ratio of 1:5 and incubated for 8 h. Luciferase activity was assayed in an Amersham Pharmacia Biotech luminometer using the Dual-Glo^®^ Luciferase Assay System (Cat#2920, Promega, Madison, WI, USA) according to the manufacturer’s instructions. Firefly luciferase activity was normalized to Renilla activity as the control for each well. 

### 2.6. Statistical Analysis

The data conforming to the normal distribution was performed by *t*-test (two-tailed, unpaired) using the Graph Pad Prism 8 program. Others was performed using a Mann-Whitney U test. All experiments were repeated at least three times independently. A *p* value of less than 0.05 was considered significant. 

## 3. Result

### 3.1. Clinical Features of the Patient

The patient was a 40-month-old male patient with growth retardation. His height was 86.9 cm (−3.2SD) and weight was 11.5 kg (−2.3SD). The boy was the first child of the first pregnancy of healthy, nonconsanguineous parents; the parents did not have any familial diseases or genetic defects. There were no medications used during the pregnancy and the mother was not exposed to cigarettes, alcohol, or drugs. The pregnancy was approximately 37 weeks in gestation. He was born by cesarean section. The birth length and birth weight of the child was 48 cm (−1.3SD) and 2300 g (−2.6SD), respectively. He showed intrauterine growth retardation. The child had thick eyebrows, narrow palpebral fissures, long and curly eyelashes, a concave nasal ridge, a long and smooth philtrum and a thin upper lip vermilion, and downturned corners of the mouth (Figure 1A). He showed abnormalities of the limbs including small hands, a short left fifth toe and a mutilated left palm (Figure 1B). Serum insulin-like growth factor 1 (IGF-1) was 71.8 nmol/L (+0.37SD) and the IGF-binding protein was 3.1 mg/L. Thyroid function and cortisol levels were normal. Echocardiography revealed a structurally and functionally normal heart. Magnetic resonance imaging of the pituitary gland was normal. A Griffiths development assessment scale—Chinese version showed an intelligence quotient of 56.45. GH stimulation tests with clonidine and arginine showed a serum GH peak of 10.77 ng/dl, making GH deficiency unlikely. Treatment with long-acting human growth hormone (PEG-rhGH) at a dosage of 0.2 mg/kg·w subcutaneously was started at 3.5 years of age. At the age of 5, the child was 103 cm (−1.9SD) tall and weighed 17.5 kg (−0.7SD). After 1.5 years of treatment, the patient’s height increased by 16.1 cm and the average annual growth rate was 10 cm/year (Figure 2).

### 3.2. Genetic Diagnosis

To determine the genetic etiology in the proband, we performed a trio-WES analysis on the patient and his parents. Peripheral blood samples were drawn from the child and his parents and genomic DNA was extracted. With an average base quality of 40 in the FASTQ files and a coverage of 99.59% in the targeted regions, the proband’s trio-WES satisfied the mutation screening criteria. A total of 50,245 variants were found in the coding and non-coding regions. The proportion of coding region variants accounted for 48.57%, including synonymous variants, missense variants, frameshift variants, non-frameshift variants, nonsense mutations, and starless (Figure 3C,D). Nonsynonymous substitutions and single nucleotide polymorphisms (SNPs) with a minor allele frequency (MAF) lower than 5% were filtered using SIFT. After screening, the number of SNPs and indels were 46,990 and 3350, respectively. We found that 933 SNPs and 346 indels were deleterious mutations according to Provean, SIFT, Polyphen2-HVAR, Polyphen2-HDIV, MutationTaster, and protein structure prediction software. The function of genes and their pathogenicity were then analyzed by referencing OMIM. Next, 592 SNPs and 182 indels were filtered out by the above procedures. Finally, through the clinical manifestation and the heredity pattern analysis, a heterozygous mutation of the *NIPBL* (c.-467C > T) was found. The SNP is located at chromosome 5 (chr: 36, 876, 893) according to the UCSC (hg37) database (http://genome.ucsc.edu/ accessed on 23 February 2022). This novel heterozygous mutation c.-467C > T of the *NIPBL* identified in the child was not present in the parents (Figure 3). This mutation is located in the 5′ UTR of the *NIPBL* and has not been reported in the literature, HGMD, gnomAD and Clinvar. According to the classification of the latest information from the ACMG (American College of Medical Genetics), it was predicted to be pathogenic.

### 3.3. Decreased Expression of the Variant Transcript

To determine the consequence of the c.-467C > T (GRCh37 chr 5: 36, 876, 893 C > T) *NIPBL* variant on gene expression, qRT-PCR was performed in the nuclear family. Peripheral blood mononuclear cells were isolated from the peripheral blood and total mRNA was extracted. The transcriptional level of the *NIPBL* variant (normalized to GADPH mRNA; mean ± SD, 0.356 ± 0.03) was significantly reduced compared with the expression levels in his father (1.054 ± 0.061) and his mother (0.94 ± 0.064) (Figure 1C). The expression level of the variant was less than half of the value of the wild-type transcript.

### 3.4. Truncations of the 5′ UTR Influence the Transcription Levels of the NIPBL

Our results indicated that the mutation in the 5′ UTR might lead to transcript downregulation. To investigate whether some critical sequences in the 5′ UTR of the *NIPBL* are implicated with its expression, various truncated 5′ UTR mutants (SPO0–SPO6) were cloned to the luciferase reporter plasmid pGL4.48 and luciferase assays were performed (Figure 4). The fragments of SPO0, SPO1, SPO3, and SPO5 contain the reference allele at c.-467. As shown in Figure 5A, SPO1 and SPO3 retained nearly 40% of their transcriptional activity compared with the full-length 5′ UTR (SPO0). By contrast, a 14.3-fold decrease in activity was observed with SPO2 and the luciferase activity of SPO4 was only 0.53%. The fragment of SPO3 was then segmented into two fragments: SPO5 and SPO6. Although the luciferase activity of SPO5 was slightly higher than that of SPO6, the regulatory effect of the two pieces was modest. Figure 5B shows the changes of the relative fluorescence values with each mutant over time. The fluorescence intensity of each truncated protamine increased gradually, 20 h after transfection, and most reached a plateau. Significant differences were observed among panel A (SPO0), panel B (SPO1 and SPO3), and panel C (SPO2, SPO4, SPO5, and SPO6).

## 4. Discussion

In this study, we report the findings of a patient with CdLS. Because of the diversity of clinical presentations of CdLS, Kline et al. recommended a scoring system that categorizes a case as classical CdLS if the patient exhibits at least three cardinal features and achieves a total clinical score equal to, or more than, 11 [21]. Our patient presented with four cardinal CdLS features, such as thick eyebrows, a concave nasal ridge, a long and smooth philtrum, and downturned corners of the mouth. Furthermore, he showed suggestive features such as intellectual disability, postnatal growth retardation, and a short fifth toe. His clinical manifestations and clinical score suggested classical CdLS. To clarify the diagnosis, trio-WES was performed on the boy and his parents. We found that the patient harbored a novel heterozygous mutation (c.-467C > T) in the *NIPBL* gene, which was predicted to be pathogenic. These findings led to enhanced skepticism as to whether the diagnosis of the patient was CdLS.

The genotype–phenotype correlation in CdLS is unclear. Both prenatal and postnatal growth retardation is a hallmark of CdLS and it is influenced by the nature of the variant and the causative gene [22]. Compared to patients with *NIPBL* mutations, individuals with *SMC1A* mutations do not show severely affected growth [23]. If the growth rate is lower than expected, gastrointestinal problems, thyroid dysfunction, and GH disorders should be considered. Most children have normal GH secretion. Our patient did not show thyroid dysfunction, and GH disorders; after the application of long-acting GH therapy, significant effects were achieved, without adverse reactions. These findings suggest that GH therapy may be effective and safe for short stature children with CdLS, similar to a previous report [23].

Gene changes affecting RNA transcription and proteins, such as frameshift mutations, have been found in patients with severe phenotypes of CdLS. In contrast, point mutations that do not affect the reading frame of proteins usually occur in patients with mild phenotypes. The clinical picture of patients with CdLS carrying *SMC1A*, *SMC3* and *RAD21* pathogenetic variants is more uniform and is characterized by a mild to moderate phenotype that is similar to the *NIPBL*-mutated probands who carry missense changes [6,24]. To date, a total of 507 *NIPBL* mutations have been reported, including missense or nonsense mutations, splicing changes, and small deletions and insertions (HGMD database). In addition, the majority of the mutation sites are located in the coding regions rather than non-coding regions, such as the 5′ UTR. Only two cases of CdLS were reported; one was caused by the c.-321-320delCCinsA variant of the *NIPBL* [18] and the other was caused by the c.1-94C > T mutation of the *NIPBL* [20]. The c.1-94C.T mutation demonstrated to be a regulatory mutation as it seems to create a new ATG site that predicts an alternative transcript and protein. The pathogenic mechanisms of the 5′ UTR variant remain unknown, and future study is required to determine how the mutations in the 5′ UTR might contribute to the occurrence of CdLS.

The 5′ UTR is located in the mRNA upstream of coding sequences and plays an essential role in gene expression [25]. Some molecules, including miRNAs, regulate mRNA abundance by binding specific sequences in the 5′ UTR [26]. We observed a substantial reduction in *NIPBL* expression in the peripheral blood of the patient harboring the c.-467C > T mutation. Dual-Luciferase reporter assays were used to identify critical regulator sequences using 5′ UTR mutants. Regrettably, we are uncertain how the variant of 5′ UTR affected *NIPBL* expression, and it could be affected by both changes in promoter activity or mRNA stability. Nevertheless, 5′ UTR mutants surely affect gene expression. Our study showed that the truncations of SPO1 (-490 bp to -360 bp) and SPO3 (-490 bp to -401 bp) induced the highest activity, and no change in the trend over time was detected. Therefore, we hypothesized that the sequences c.-490 to -401 in the 5′ UTR of the *NIPBL* might play a decisive role in transcription control and regulation of gene expression. It should be noted that this study has examined only sections of sequences and has not identified binding sites of regulatory molecules.

In conclusion, we found a novel heterozygous mutation (c.-467C > T) in the *NIPBL* gene resulting in CdLS, providing strong evidence for a definitive diagnosis in this patient. This result further expanded the spectrum of pathogenic mutations for CdLS. Our experiments in vitro have begun to elucidate the important regulatory sequences in the 5′ UTR of the *NIPBL*, laying the groundwork for further study.

## Figures and Tables

**Figure 1 genes-13-00740-f001:**
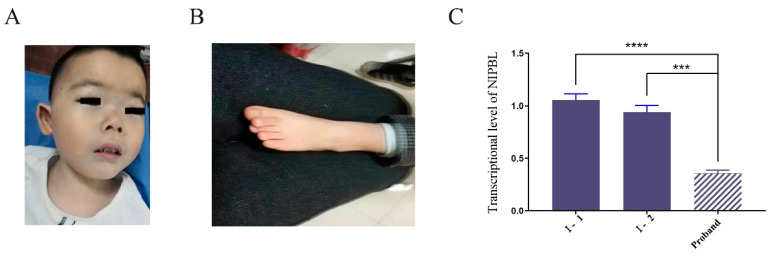
Clinical features. (**A**) The patient had well-defined and arched eyebrows with synophrys; narrow palpebral fissures; long and curly eyelashes; broad nose with high nasal bridge; and broad mouth. (**B**) The short left fifth toe was observed in the boy. (**C**) Quantitative analysis of *NIPBL* expression in patient and his parents. *** *p* ≤ 0.001, **** *p* ≤ 0.0001.

**Figure 2 genes-13-00740-f002:**
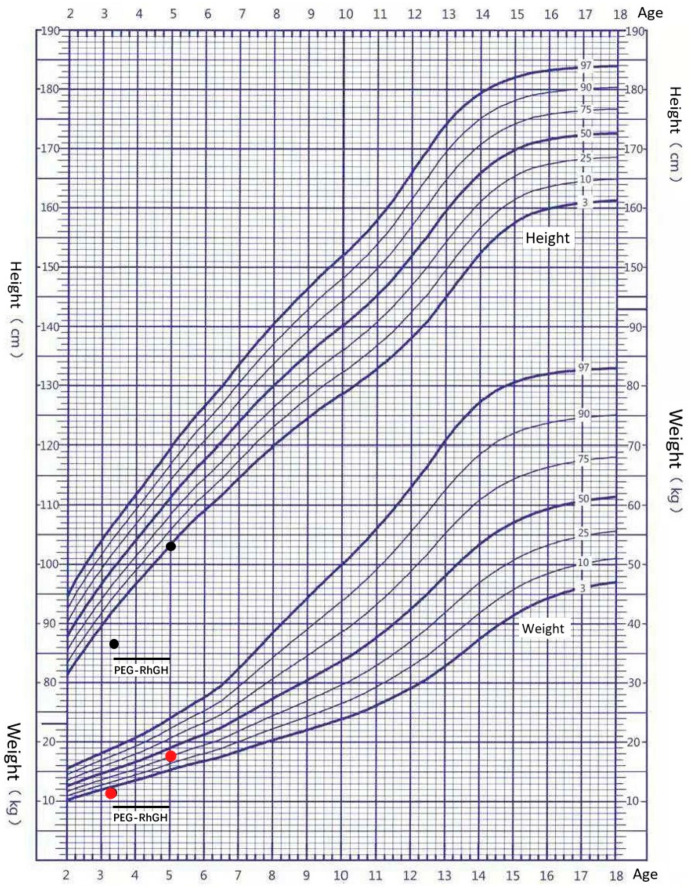
Growth curve of boys aged 2–18 years in China. The black dots represent changes in height after (PEG-rhGH) treatment. The red dots represent changes in weight after (PEG-rhGH) treatment.

**Figure 3 genes-13-00740-f003:**
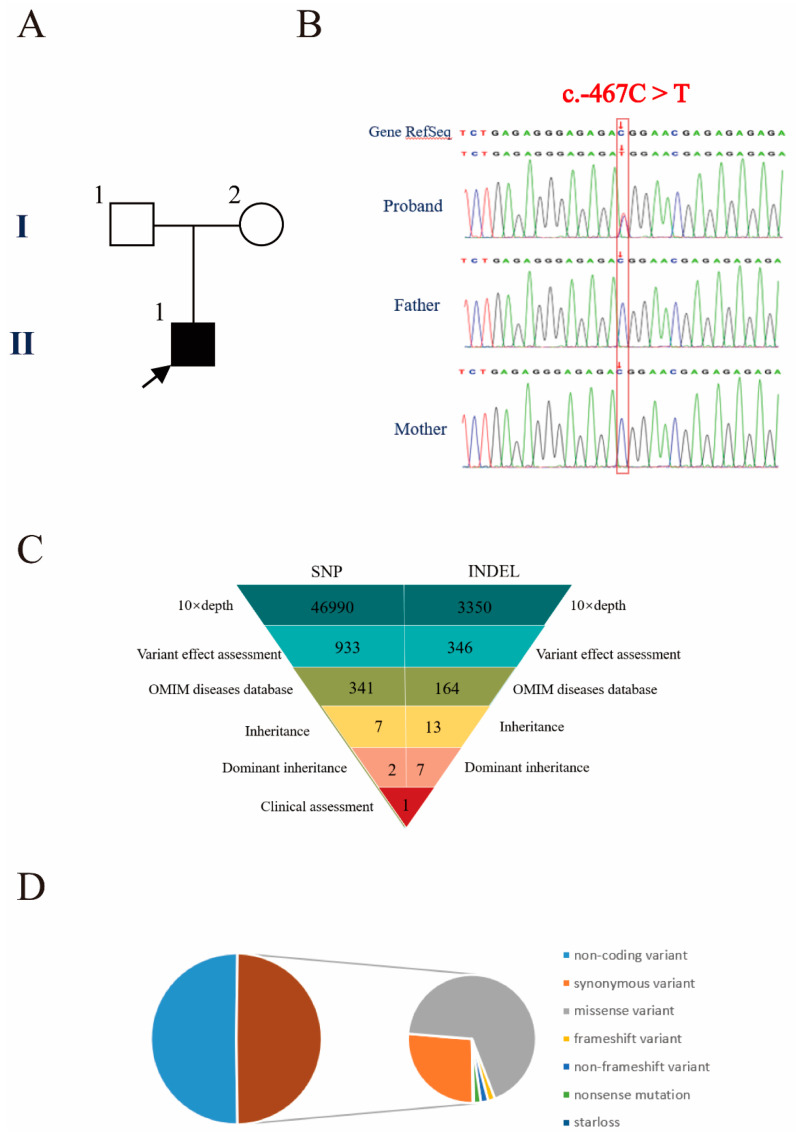
Genetic diagnosis. (**A**) Pedigree of the family with segregation of the identified *NIPBL* mutation. The square and circle represent the male and female respectively, and the arrow indicates the proband. A filled symbol represents a person affected with CdLS. (**B**) Results of gene sequencing show that mutation from affected individuals and wild type from unaffected family members. (**C**) Schematic representation of the exome data filtering approach under the assumption of dominant inheritance in the family. (**D**) Percentage of variants in exome regions.

**Figure 4 genes-13-00740-f004:**
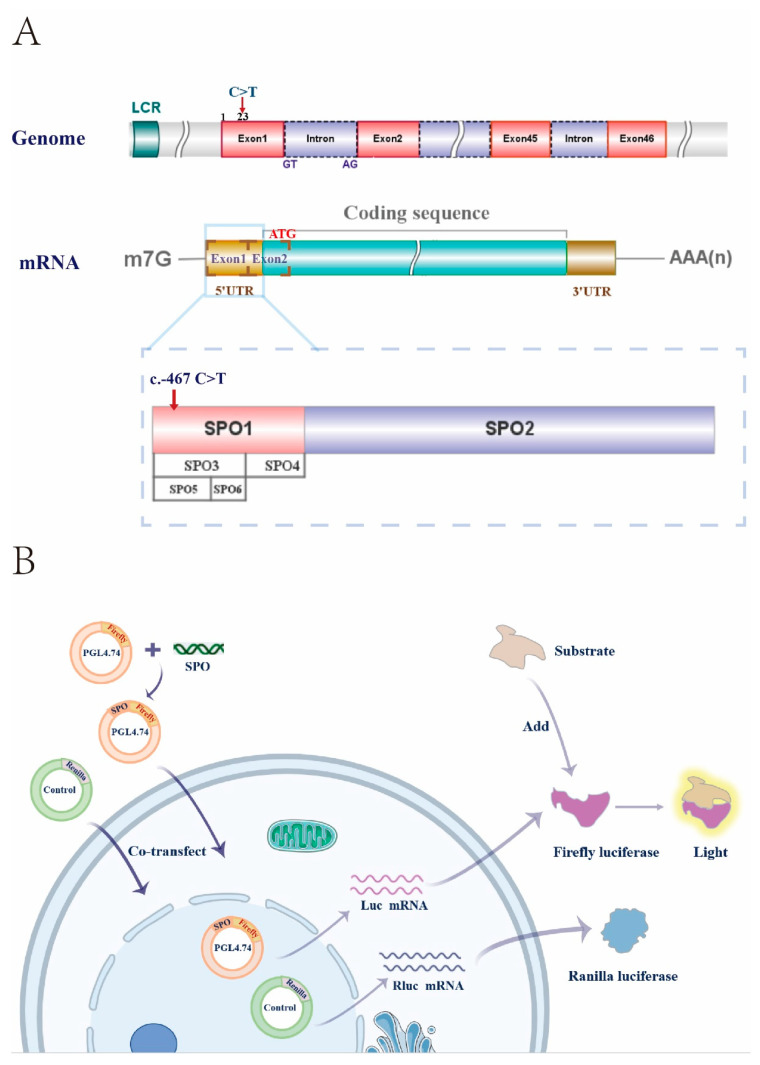
Schematic of principle. (**A**) Schematic representation of the *NIPBL* 5′ UTR engineering strategy. (**B**) The schematic of Dual-Luciferase reporter gene assay system.

**Figure 5 genes-13-00740-f005:**
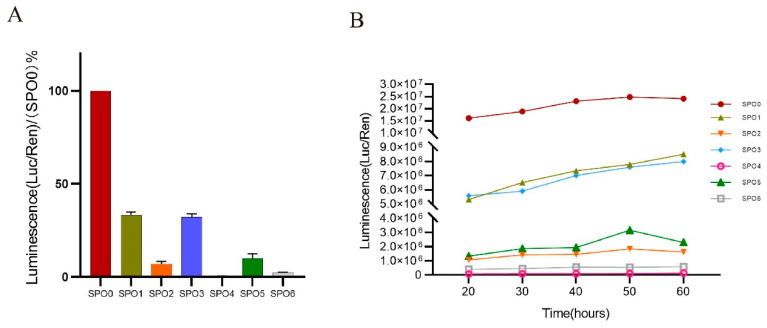
Transcription activity of 5′ UTR truncations. (**A**) Relative luciferase activity obtained from the different constructs. SPO1 to SPO6 are located in 5′ UTR. SPO1 (c.-490 to c.-360), SPO2 (c.-360 to c.-1), SPO3 (c.-490 to c.-401), SPO4 (c.-401 to -360), SPO5 (c.-490 to c.-441), and SPO6 (c.-441 to c.-401), respectively, (ATG start codon = +1). (**B**) Luciferase values of different constructs every 10 h since twenty hours after transfection.

## Data Availability

All datasets generated for this study are included in the article.

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
