# Peer review of "A Novel de Novo Variant in 5′ UTR of the NIPBL Associated with Cornelia de Lange Syndrome"

_genes, 2022, doi:10.3390/genes13050740_

Round 1

Reviewer 1 Report

The authors present their data of a novel mutation in 5`UTR in the NIPBL gene associated with Cornelia de Lange syndrome. They completed the clinical case report by adding in vitro research.

This is an interesting study and the analyses seem mostly appropriate. However, there are some topics/results that are not discussed/obtained and some major remarks.

Major comments

  • general: it should be stated in the abstract and text that it is a de novo variant!
  • abstract: “NIPBL variants occur in approximately 70% of CdLS cases.”, this is rather 80% (worldwide) instead of 70%, or mention the population regarding 70%.
  • introduction: 7 genes, I would rather say 6; ANKRD11 has been linked to KBG syndrome, rather than CDLS, although there could be some clinical overlap. I would not relate this gene to CDLS and rather see it as a differential diagnosis.
  • abstract: it should be stated somewhere that 5UTR mutations in NIPBL have already been linked to CDLS
  • discussion: “Only two cases of CdLS were reported; one was caused 269 by the c.-321-320delCCinsA variant of NIPBL 18” how about reference 20?? please elaborate
  • discussion: “in the NIPBL gene, which was predicted to be pathogenic.” I am curious what the initial genetic rapport said? was it a variance of unknown significance (VUS) or what? please mention the initial finding throughout the manuscript
  • discussion: incorrect?: “Our patient did not show any of the above issues”=> however your patient had growth retardation, so this is an aforementioned “issue”. please correct text?
  • (!) Recommendation: it would be nice to make an overview of the currently reported NIBPL gene mutations (and/or regions; eg similar to figure 3 https://pubmed.ncbi.nlm.nih.gov/33144681/ ), linked with CDLS. => like stated in the text “total of 507 NIPBL mutations have been reported, including missense or nonsense 266
  • mutations, splicing changes, and small deletions and insertions (HGMD database”
  • section 3.1: was there already intrauterine growth retardation?
  • figure: “(B) The short left fifth finger was observed in the boy” this picture is from the foot? do you mean the toe is short or??? please adjust text and/or figure…
  • “ A total of 50245 variants were found in coding and non-coding regions.” non-coding regions? so you performed WGS? WES (whole exome sequencing) will not find non-coding region variants?! please adjust
  • “This mutation is located in the 5′UTR of NIPBL and has not been reported in the literature.” okay, and has it been reported in gnomAD or other population databases? please mention…
  • figure 2B: I think there is a mistake : in the figure “-467C>G” should be “c.-467C>T”?!
  • figure 4: please mention the abbreviations used in your figure by full text and guide the reader with some brief explanation…it is obligatory to understand the figure without reading the text, which is not the case now… eg SPO1 (-490 bp to -360 bp) and SPO3 (-490 bp to -401 bp)

minor comments

  • ref 18,20 in red line 59
  • 1: the font of the text is different, please adjust
  • why is some text in red? please adjust….
  • statistics: “All the statistical analysis was performed by t test” are you sure that the data were normally distributed? if not, a MWU-test seems more appropriate… Please mention if and how you analyzed the normal distribution of the data (e.g. D’Agostino and Pearson omnibus normality test??).

Reviewer 2 Report

The study by Chen and colleagues presents the genetic research of a patient with rear syndrome: Cornelia de Lange. The Introduction,  Materials and Methods sections are very well presented.  
Overall the topic of the study is not particularly original. Its relevance mainly lies in providing evidence of novel mutation in 5'UTR of NIPBL in the child with Cornelia de Lange syndrome. It expends the spectrum of known pathogenic mutations for this syndrome. Analyses are appropriately conducted, and relevant results are reported and correctly interpreted. Conclusions are adequately summarised. The quality of the figures is overall satisfactory. The reference list is updated and covers relevant literature.
There is nothing to improve in the presented manuscript. 

Author Response

Thank you for your affirmation of this research, and we will continue to work hard.

Round 2

Reviewer 1 Report

Can be accepted after a final spelling check.

Great work!

Author Response

Thank you very much for the careful review and valuable suggestions concerning our manuscript(1631018).English expression has been carefully improved throughout the manuscript by professional company.And the company helps revise it again.Thank you very much for considering our manuscript. We are looking forward to your response. If you have any questions, please do not hesitate to contact me.